



# Version 8 IMK/IAA MIPAS ozone profiles: nominal observation mode

Michael Kiefer[1], Thomas von Clarmann[1], Bernd Funke[2], Maya García-Comas[2], Norbert Glatthor[1], Udo Grabowski[1], Michael Höpfner[1], Sylvia Kellmann[1], Alexandra Laeng[1], Andrea Linden[1], Manuel López-Puertas[2], and Gabriele P. Stiller[1]

[1]Karlsruhe Institute of Technology, Institute of Meteorology and Climate Research, Karlsruhe, Germany
[2]Instituto de Astrofísica de Andalucía, CSIC, Granada, Spain

**Correspondence:** Michael Kiefer (michael.kiefer@kit.edu)

**Abstract.**

A new global $O_3$ data product retrieved from Michelson Interferometer for Passive Atmospheric Sounding (MIPAS) spectra with the IMK/IAA MIPAS data processor has been released. These data are based on ESA version 8 recalibration of radiance spectra which takes detector aging into consideration to minimize drifts. The new ozone retrievals use improved temperatures and thus suffer less from the propagation of related errors. Changes in the level-2 processing with respect to previous data versions and relevant to ozone include: (1) The background continuum is now considered up to 58,km. (2) A priori information is now used to constrain the retrieval of the radiance offset. (3) Water vapour is jointly retrieved along with ozone mixing ratios. (4) A more adequate regularization has been chosen. (5) Ozone lines in the MIPAS A band (685–980 cm$^{-1}$) are used almost exclusively because of inconsistencies in spectroscopic data of the MIPAS AB band (1010–1180 cm$^{-1}$). Only at altitudes above 50 km, where A band ozone lines do not provide sufficient information, ozone lines in the MIPAS AB band are used. (6) Temperature-adjusted climatologies of vibrational temperatures of $O_3$ and $CO_2$ are considered to account for non-local thermodynamic equilibrium radiation. Ozone errors are estimated to be less than 10% in the altitude range 20–50 km. The error budget is dominated by the spectroscopic errors of ozone and carbon dioxide. The latter error contribution is propagated from the results of temperature and line-of-sight retrievals. Further notable contributions are the uncertainty of the instrumental line shape function, the gain calibration error, and the spectral noise, directly in the ozone lines and propagated via the previously retrieved temperature and line-of-sight. The error contribution of interfering gases is almost negligible. The vertical resolution in terms of the full width at half maximum of the averaging kernel rows depends on altitude and atmospheric conditions. For the measurement period 2002–2004 it varies between 2.5 km at the lowest altitudes and 6 km at 70 km, while in 2005–2012 it covers 2 to 5.5 km in the same altitude range. The horizontal smearing in terms of the full width at half maximum of the horizontal component of the 2-dimensional averaging kernel matrix is smaller than, or approximately equal to, the distance between two subsequent limb scans, at all altitudes. This implies that the horizontal resolution is sampling-limited or optimal, respectively. Along with the regular representation of the data, that have non-unity averaging kernels, a resampled data version is made available that is free of formal a priori information and thus more user-friendly for certain applications. Version 8 ozone results show a better consistency between the two MIPAS measurement periods. They seem to be more realistic than





preceding data versions in terms of long-term stability, as at least a part of the drift is corrected. Further, the representation of elevated stratopause situations is improved, but there is still some indication of a positive bias in the upper stratosphere.

# 1  Introduction

The limb-viewing Michelson Interferometer for Passive Atmospheric Sounding (MIPAS) was operational from June 2002 to April 2012. The mid-infrared Fourier transform spectrometer recorded atmospheric emission spectra with a spectral coverage
of 4.1 to 14.7 $\mu$m (685–2410 cm$^{-1}$). The polar sun-synchronous orbit allowed global coverage of measurements (Fischer et al., 2008). This paper describes the retrieval of ozone with a data processor developed and operated by the Institute of Meteorology and Climate Research (KIT-IMK) in cooperation with the Instituto de Astrofísica de Andalucía (IAA-CSIC), based on the most recent level-1b data version provided by the European Space Agency (ESA version 8.03). The characteristics of these radiance spectra are discussed in Kiefer et al. (2021).

The retrievals presented here include measurements recorded during two operational phases. In the first phase MIPAS measured with full spectral resolution (full resolution, FR) of 0.025 cm$^{-1}$ (unapodized). After a defect of the interferometer slide, in the second phase of the mission MIPAS measured with reduced spectral resolution of 0.0625 cm$^{-1}$ (reduced resolution, RR). The respective ozone data versions are: V8_O3_61 for FR nominal mode (NOM) observations, V8_O3_261, for RR NOM observations, and V8_O3_161 for measurements in the observation mod ededicated to upper troposphere and lower
stratosphere measurements (UTLS-1).

The IMK/IAA MIPAS data processor has been designed to retrieve global temperature and composition distributions from the level-1b data. In this paper we first summarize the effects of the improved calibration (Section 2) and then present improvements in the retrieval setup of ozone compared to earlier data versions. Finally we discuss the impact of these retrieval settings on the resulting ozone distributions (Section 3).

The fact that a formal constraint is implemented with the retrieval adds complications to the correct use of the data. It implies that the best use of the data can only be made by the correct application of averaging kernels in order not to misinterpret the data. Further, the variable vertical resolution of the ozone profiles can make applications like, e.g., trend assessment, problematic. To avoid these problems, for the first time a maximum likelihood representation is provided along with the regular representation of the data, in order to provide an alternative to users who prefer data free of formal prior information (von Clarmann et al.,
2015, and references therein). The characteristics of this representation are summarized in Section 6.

In Section 7 we assess to which degree the changes inherent in the new data version were successful. We review which of the known drawbacks of previous data versions have been reduced, or even removed.

## 1.1  Lessons learned from previous studies

Previous MIPAS data versions proved to be a useful and reliable data set and were widely used by the scientific community
(Funke et al., 2011; Tegtmeier et al., 2013; Sofieva et al., 2013, 2017; Funke et al., 2017). Still several issues with these data



versions were detected. These are summarized in the following, because to remedy them was the chief motivation to provide still another data version.

Eckert et al. (2014) found ozone drifts in MIPAS version 5 ozone data of up to 0.3 ppmv/decade, depending on latitude and altitude. These were mainly negative and attributed to an inadequate treatment of the detector non-linearity. Laeng et al. (2018) provided evidence that the negative drift was reduced in data version 7.

MIPAS IMK/IAA data version 5 had a positive bias in the upper troposphere and lowermost stratosphere (Laeng et al., 2014, 2015). This bias was significantly reduced in version 7. It was found that the use of microwindows in the MIPAS AB band (1010–1180 $cm^{-1}$ in level-1b version 8; 1020-1170 $cm^{-1}$ in older versions) in data version V5R_O3_222 caused a positive bias in the ozone maximum and slightly displaced the peak (Laeng et al., 2014). Inconsistencies between spectroscopic ozone data at different mid-infrared wavelengths were already suspected by Glatthor et al. (2006) but no final conclusion could be drawn then because MIPAS inter-band calibration inconsistencies were another candidate explanation for the positive bias. A more recent study (Glatthor et al., 2018) has corroborated the idea of inconsistent spectroscopic data. Beyond this spectroscopic inconsistency problem, the AB band microwindows were more susceptible to the nonlinearity/drift issue discussed above. In V5R_O3_222, the AB band microwindows were thus discarded. To compensate for the loss of information implied by dropping the AB microwindows at heights below 50 km, three-times-more microwindows were used in the A band in this height range. This improved the previously poor vertical resolution around the ozone vmr maximum. However, without the AB band microwindows, less information about ozone in mesospheric altitudes (above about 50 km) is obtained. In V5R_O3_224, the AB band microwindows were included again, however, at altitudes above 50 km only. The rationale behind this decision was that at these altitudes the gain in precision by inclusion of these AB microwindows outweighed the induced bias.

The goal of the version 8 processing is to combine the most recent developments in the retrieval strategy with the best available, namely version 8, set of calibrated level-1b spectra in order to provide the best possible ozone data set.

## 2 Calibration

MIPAS time series of ozone data retrieved with the IMK/IAA research processor were shown to be affected by drifts (Eckert et al., 2014). Also, drifts were detected for the ESA operational ozone product (Hubert et al., 2016). As the cause of these drifts an inadequate non-linearity correction in the gain calibration of previous level-1b data versions was identified. The preflight measurements of detector non-linearity used for the calibration of MIPAS spectra up to version 5 were no longer applicable since the non-linearity of the detector response was found to be time-dependent. Due to aging, the detector response became more linear with time (Birk and Wagner, 2010). These authors have developed a correction scheme that considers the age-dependence of the detector-nonlinearity. It has been shown for version 7 that this new calibration tool removes at least a major part of the drift in the ozone time series of version 5 (Laeng et al., 2018). The in-flight characterization of the detector nonlinearity has been further improved in version 8 spectra (Kleinert et al., 2018). The calibration-related changes since previous data versions are described in Kiefer et al. (2021).



## 3    Retrieval

In this paper, the most recent ozone retrievals performed with the IMK/IAA MIPAS data processer are discussed. This data
processor uses a constrained multi-parameter non-linear least squares fitting scheme (von Clarmann et al., 2003). It has been
extended to allow retrievals under consideration of non-local thermodynamic equilibrium emissions (Funke et al., 2001). von
Clarmann et al. (2009b) describe its application to MIPAS RR measurements. The underlying mathematics of the retrieval of
MIPAS version 8 ozone data is discussed at length in Kiefer et al. (2021). The retrieval of ozone mixing ratio profiles follows
the retrieval of temperature and precedes in the retrieval chain the retrieval of water vapour.

The profile of ozone volume mixing ratios retrieved from FR and RR measurements is represented on a discrete retrieval
grid with a lowest point at 0 km, then there is a gridwidth of 1 km between 4 and 55 km and further gridpoints at 56.5, 58, 60,
62, 64, 66, 68, 70, 72.5, 75, 77.5, 80, 85, 90, 95, 100, 105, 110, and 120 km.

Here we report only the retrieval settings and their changes since the last data version which are specific to the ozone retrieval.
These are: (a) the use of version 8 temperatures and tangent altitude information (Section 3.1), (b) the retrieval of a background
continuum up to 58 km (Section 3.2.1), (c) the use of prior information for the radiance offset retrieval (Section 3.2.2), (d) the
joint retrieval of water vapour mixing ratios (Section 3.2.3), (e) a more adequate regularization (Section 3.3), (f) the use of
additional microwindows (Section 3.4), (g) use of carefully selected spectroscopic data (Section 3.5), and (h) the operation of
the radiative transfer model at higher numerical accuracy (Section 3.6).

### 3.1    The preceding temperature retrieval

For the ozone retrieval, MIPAS temperatures and tangent altitude information is used which are provided by preceding re-
trievals. A major part of the improvement of the new ozone product is attributed to more reliable temperatures, particularly in
elevated stratopause situations. These are relevant above the uppermost tangent altitude where MIPAS does not vertically re-
solve the temperature profile and hence the retrieval is particularly sensitive to the shape of the a priori temperature profile. The
climatological mesospheric profiles used in previous retrieval versions do not represent these situations well, and the related
temperature errors in the upper part of the temperature profile propagate to lower layers, and propagate further onto the ozone
profiles. This problem could be remedied by the use of bias-corrected temperature profiles from a specified dynamics run (Gar-
cía et al., 2017) of the Whole Atmosphere Community Climate Model (WACCM, Marsh, 2011; Marsh et al., 2013) Version 4
(WACCM4). The model output better represented exceptional atmospheric conditions like elevated tropopause events (Kiefer
et al., 2021).

All changes implied by the revision of the temperature retrieval map onto the ozone results. The new treatment of tempera-
tures above the highest MIPAS tangent altitude has major implications on the ozone retrievals. In particular, the hitherto existing
bias in the overlapping altitude range between nominal measurements and measurements taken with middle atmosphere mea-
surement scenarios (MA), the latter reaching higher up in the atmosphere, has disappeared. There is no more need to further
homogenize the data set. This indicates that temperature uncertainties above the highest tangent altitude of the nominal MIPAS
observation scenario were among the causes of ozone inconsistencies between the NOM and MA mode.





Other improvements in the retrieval of temperature distributions reported by these authors are related to regularization, the treatment of interfering species, $CO_2$ mixing ratios, the horizontal structure of the atmosphere, the selection of microwindows, the treatment of the background continuum and the radiance offset, spectroscopic data and numerical issues.

The temperatures entering the ozone retrieval forward calculations are the 3D temperature field discussed in Section 3.5 of Kiefer et al. (2021). This means that the horizontal temperature structure is based on ECMWF ERA-Interim data, extended by NRLMSISE-00 data above 60 km, and that the vertical structure is based on the result of the temperature retrieval (local scaling).

## 3.2 The unknowns of the retrieval

The target quantities of the retrieval are ozone volume mixing ratio, water vapour mixing ratio, background continuum, and radiance offset. The latter two are retrieved independently per microwindow.

Tests were performed to include the horizontal gradient of ozone as a further unknown. However, neither did the resulting vertical profiles of the gradients bear much resemblance with the horizontal gradient calculated directly from adjacent ozone vmr profiles, nor was there any improvement in the $\chi^2$-values of the retrieval. Therefore, this option was not pursued any further, and not used in the final retrieval setup.

### 3.2.1 Background continuum

In order to account for all weakly wavenumber-dependent radiance contributions which result neither from line-by-line calculations of absorption cross-sections nor from consideration of interpolated pre-tabulated cross-section spectra of heavy molecules, a background continuum is fitted to minimize the residual between measured and modelled spectra (von Clarmann et al., 2003). The continuum is described by an absorption cross-section which is constant within a microwindow but can vary between microwindows and with altitude. For the reasons discussed in Kiefer et al. (2021), the background continuum has now been considered at altitudes up to 58 km, as opposed to 48 km in version 5.

### 3.2.2 Radiance offset

The radiance offset is an additive term included to correct the radiance zero-level calibration. It is constant within a microwindow. A priori infomation on the offset was taken from Kleinert et al. (2018) and was necessary to be included because at higher altitudes, where radiative transfer is linear, the background continuum and the radiance offset would form a null space in the retrieval.

### 3.2.3 Water vapour

Ozone spectral lines used for the retrieval are appreciably interfered with water vapour spectral lines. In order to avoid propagation of assumptions on the water vapour profiles on the ozone data product, water vapour is fitted jointly with ozone. Since





the resulting water vapour profiles are inferior to the regular MIPAS H₂O product, the water vapour information resulting from the ozone retrievals is not considered any further.

### 3.3 Regularization

Regularization of the V8 ozone retrieval uses a smoothing term based on a squared first order difference based cost function (see, e.g., Tikhonov, 1963; Twomey, 1963; Phillips, 1962). The a priori profile is a constant profile (a zero profile). Additionally, a diagonal term is used, which pushes the result towards the a priori profile, similar as optimal estimation or maximum a posteriori retrievals (Rodgers, 2000). This diagonal term is employed only at the two uppermost altitudes of the retrieval grid. Values are pushed towards zero vmr there. The purpose is to avoid unphysical large negative mixing ratios at these altitudes which are occasionally retrieved otherwise. This modification allowed to weaken the Tikhonov-type smoothing regularization at all altitudes by approximately 20%. The approach to control the altitude dependence of the smoothing constraint as described by Kiefer et al. (2021, their Eq. 3) supersedes the old approach by Steck and von Clarmann (2001) which was previously used.

### 3.4 Microwindows

In a former MIPAS data version (V5R_O3_224) a positive bias in MIPAS ozone profiles in the upper troposphere and lowermost stratosphere was detected (Laeng et al., 2014, 2015). This problem could be attributed to the use of microwindows in the MIPAS AB band (1010–1180 cm$^{-1}$). Glatthor et al. (2018) showed that ozone spectroscopic data in the MIPAS A band (685–980 cm$^{-1}$) and the AB band were inconsistent. While removal of the AB band microwindows removed the ozone bias, insufficient measurement information was available at higher altitudes, leading to increased retrieval noise and less than optimal vertical resolution. For ozone version 8 some AB band microwindows were re-included, but only at altitudes above 50 km, where the gain in precision and vertical resolution outweighs the bias caused by the microwindows. The version 8 microwindow selection is shown in Table 1. The microwindow 720.7500 cm$^{-1}$ to 723.6875 cm$^{-1}$, which was still included in ozone data version 7, to which the evaluation by Laeng et al. (2018) refers, has been removed, because of systematic fit residuals that could not be removed.

### 3.5 Spectroscopic data

For O₃ and HNO₃, the MIPAS spectroscopy data sets pf3.2 and pf4.45 (Flaud et al., 2003a,b) are used, respectively. For the other interfering species HITRAN 2016 (Gordon et al., 2017) is used. CO₂ line mixing coefficients have been re-calculated for the new spectroscopic data. The choice of the MIPAS spectroscopic ozone database was motivated by the fact that the inconsistencies between the MIPAS A and AB bands are smaller than with the HITRAN data and because in HITRAN 2016 there is an implausible jump in the air broadening coefficients at 797 cm$^{-1}$ (Glatthor et al., 2018).





**Table 1.** Microwindows used in the MIPAS NOM $O_3$ retrieval

| Wavenumber range (FR) (cm$^{-1}$) | Wavenumber range (RR) (cm$^{-1}$) | Altitude range (km) |
|---|---|---|
| 687.7000–688.6750 | 687.6875–688.6875 | 33–75* |
| 689.3250–691.8750 | 689.3125–691.8750 | 24–75* |
| 692.2500–695.1750 | 692.2500–695.1875 | 36–75 |
| 707.1250–710.0500 | 707.1250–710.0625 | 27–75 |
| 712.3250–713.4250 | 712.3125–713.4375 | 6–54* |
| 713.5000–716.4250 | 713.5000–716.4375 | 6–75* |
| 716.5000–719.4250 | 716.5000–719.4375 | 9–75* |
| 728.5000–729.3750 | 728.5000–729.3750 | 6–75* |
| 730.0750–730.5000 | 730.0625–730.5000 | 9–75* |
| 731.9500–732.8750 | 731.9375–732.8750 | 6–75* |
| 734.0000–734.7500 | 734.0000–734.7500 | 6–75* |
| 736.4500–739.3750 | 736.4375–739.3750 | 6–75* |
| 739.4500–741.9250 | 739.4375–741.9375 | 6–75* |
| 745.2500–745.6750 | 745.2500–745.6875 | 6–75* |
| 746.7000–747.1250 | 746.6875–747.1250 | 6–75* |
| 747.6250–748.3750 | 747.6250–748.3750 | 6–75* |
| 749.5750–752.5000 | 749.5625–752.5000 | 6–75* |
| 752.9500–755.8750 | 752.9375–755.8750 | 18–75* |
| 758.3750–759.4250 | 758.3750–759.4375 | 15–75* |
| 759.5000–761.8750 | 759.5000–761.8750 | 6–75* |
| 765.0000–765.6250 | 765.0000–765.6250 | 6–75* |
| 767.5000–768.0000 | 767.5000–768.0000 | 9–75* |
| 771.8750–772.1250 | 771.8750–772.1250 | 6–75* |
| 774.2500–774.5500 | 774.2500–774.5625 | 6–69* |
| 776.5000–776.7500 | 776.5000–776.7500 | 6–69* |
| 780.2500–781.9250 | 780.2500–781.9375 | 6–75* |
| 788.9500–789.6750 | 788.9375–789.6875 | 6–75* |
| 790.7500–791.0000 | 790.7500–791.0000 | 6–48* |
| 791.2000–791.5500 | 791.1875–791.5625 | 6–75 |
| 808.2000–808.7500 | 808.1875–808.7500 | 6–24 |
| 825.1250–825.4250 | 825.1250–825.4375 | 6–24 |
| 827.3750–827.8000 | 827.3750–827.8125 | 6–24 |
| 1029.0000–1031.0000 | 1029.0000–1031.0000 | 51–75 |
| 1038.0000–1039.0000 | 1038.0000–1039.0000 | 51–75 |

*The altitude range of this microwindow is the envelope range. i.e. the microwindow is not used in the entire altitude range reported but there are certain altitudes where it is disregarded to avoid the signal of interfering species of uncertain concentration.





## 3.6 Numerical issues

The radiative transfer model KOPRA (Stiller, 2000) used to provide the modelled spectra and the Jacobians was operated
180 at higher numerical accuracy than in previous versions. In particular, the monochromatic grid on which the absorption cross
sections are calculated was set to $0.0009765625$ cm$^{-1}$ instead of $0.00125$ cm$^{-1}$, and the apodization of calculated spectra
used a wider frequency range. To avoid the result flipping back and forth between subsequent iterations, the inversion program
used a specific oscillation detection scheme (see Section 3.12 of Kiefer et al., 2021). A total of 2,393,767 ozone profiles were
obtained (506,262 for FR and 1,887,033 for RR mode). The convergence rate was 99.96%.

185 ## 3.7 Non-LTE

In the upper part of the altitude range under consideration (above 60 km), the assumption of local thermodynamic equilibrium
(LTE) is not valid in the 10 $\mu$m spectral region (López-Puertas and Taylor, 2001). To avoid propagation of related effects on the
retrieved ozone mixing ratios, deviations of the populations of vibrationally excited molecules from a Boltzmann distribution
at the local kinetic temperature are considered in the radiative transfer calculations for the 010, 001, and 100 bands of the
190 ozone main isotopologue. The relatively small magnitude of these non-LTE effects in the altitude range covered by the NOM
observations does not justify the effort to employ computationally expensive non-LTE population models during each step of
the retrieval, as done previously in retrievals from observations taken in the middle and upper atmospheric observation mode
(López-Puertas et al., 2018).

Instead we use a parameterised non-LTE approach that accounts for the temperature dependence of the vibrational non-LTE
195 populations. It is based on a seasonal and latitudinal climatology for the local times of MIPAS ascending and descending
overpasses which has been computed with an updated version of the Generic RAdiative traNsfer AnD non-LTE population
algorithm (GRANADA, Funke et al., 2012). More detail on this approach can be found in Section 3.11 of Kiefer et al. (2021).

## 4 Error budget

Since thorough error calculations are computationally expensive, the estimation for the error contributions measurement and
200 instrument characterization errors, and parameter errors and model errors, is performed for a set of 34 representative atmo-
spheres (defined in Tables A1 and A2 of von Clarmann et al., 2022). For each of these 34 atmospheres a set of approximately
30 geolocations are selected and the error calculations are performed. Means are then calculated from the respective geoloca-
tions and the diverse error components. These means are regarded as error component estimates for the specific representative
atmosphere. The only exception to this approach is the noise error estimate, which is available for every single ozone profile,
since it is calculated from the spectral noise $S_y$ during the retrieval. Every single measurement geolocation can be associated
uniquely with one of those representative atmospheres. Hence, the respective error estimates, except for the noise error, are
assigned to this geolocation, with the relative errors being scaled to the actual ozone profile. Technical details of the error
estimation of MIPAS trace gas retrievals are reported by von Clarmann et al. (2022).





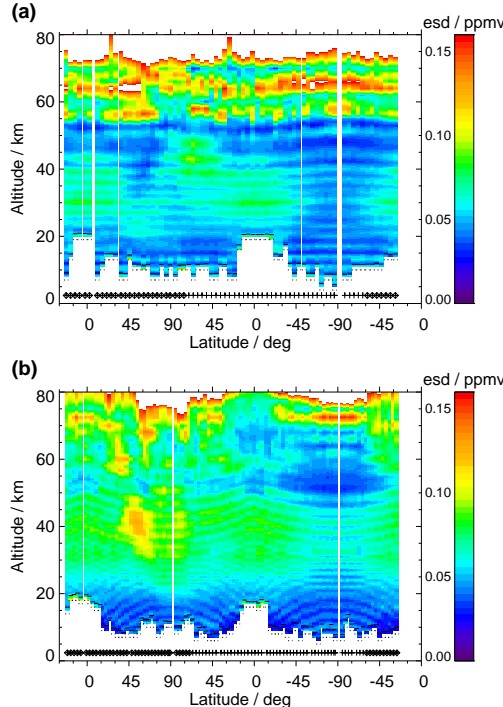

**Figure 1.** The noise-component of the ozone uncertainty in terms of estimated standard deviations for sample orbits 4577 (a) and 35800 (b) as a function of latitude and altitude. Data are shown along one orbit starting at approx. 30° S on the ascending orbit part. The symbols just above the latitude axis show the position of the measurement and indicate the illumination conditions (diamond: nighttime; plus: daytime).

The following error sources are considered: Measurement errors including spectral noise, uncertain gain calibration, in-
strument line shape uncertainty, uncertainties in the frequency calibration and pointing errors; parameter errors including temperature and mixing ratios of interfering species; and uncertainties in spectroscopic data.

As justified in detail by von Clarmann (2014) the error budget does not comprise the smoothing error: Our retrieval should be considered as an estimate of the smoothed true profile rather than a smoothed estimate of the true profile (see Rodgers, 2000, Section 3.2.1, for a discussion of this issue).

The measurement noise is typically $30–33\,\mathrm{nW/(cm^2\,sr\,cm^{-1})}$ in the MIPAS A band and $5.4–9.6\,\mathrm{nW/(cm^2\,sr\,cm^{-1})}$ in the AB band; both values refer to apodized spectra. The response of the ozone retrieval to measurement noise is shown for Envisat orbits 4577 (FR) and 35800 (RR) in panels (a) and (b) of Fig. 1, respectively. The region of increased noise around altitudes of approximately $40\,\mathrm{km}$ and around 60°N in Fig. 1b is caused by very low temperatures (less than $200\,\mathrm{K}$) which lead to reduced atmospheric signal in the corresponding IR spectra, while the spectral noise contributions are unchanged. The noise error can
roughly be considered as additive. That is to say, the relative contribution of noise is larger for smaller mixing ratios. Further, the signal to noise ratio is smaller for colder atmospheres.

**Figure 2.** Ozone error budget for FR (a, c, e) and RR (b, d, f) data. Additive and multiplicative errors are shown as relative errors of the respective ozone profiles. All error estimates are 1-$\sigma$ uncertainties. Error contributions are marked "T+LOS" for the propagated error from the T+LOS retrieval, "noise" for error due to spectral noise, "spectro" for spectroscopic error, "gain" for gain calibration error (MIPAS band A and AB, see text), "shift" for spectral shift error (see text), "ILS" for instrument line shape error (see text), "offset" for error due to spectral offset (see text), and "interf" for the error due to interfering gases. (a, b) northern midlatitude summer day, (c, d) tropical day, and (e, f) southern polar winter night.



Gain uncertainties were estimated at 1.4% during the FR period and 1.1% during the RR period for the A band; for the AB band we used 0.9% for the FR period and 0.8% for the RR period (Kleinert et al., 2018). The response of retrieved ozone abundances to the gain calibration error is roughly multiplicative. In MIPAS band A the gain uncertainties of the ozone retrieval
interact in a systematic, mostly compensating, way with gain uncertainty components of the temperature and tangent altitude errors.

Instrument line shape errors, and spectral shift residual error are treated as described in Kiefer et al. (2021). The uncertainty of the spectral shift is $0.00029\,\mathrm{cm}^{-1}$. ILS uncertainties are based on the estimates of modulation loss through self-apodization and its uncertainties (Hase, 2003).

For the ozone retrieval, information from the preceding temperature and tangent altitude retrieval is used. The temperature uncertainties can reach up to 2.5 K in few cases, but are mostly below 1 K in the altitude range 20–50 km (Kiefer et al., 2021). Tangent altitude uncertainties are in the range of 200–250 m for the systematic and 30–80 m for the random component. The latter is propagated onto the ozone product using the corresponding error covariance matrices of the temperature and line-of-sight retrieval. The resulting ozone uncertainties roughly scale with the ozone mixing ratio and should thus be treated as
multiplicative errors. The gain, ILS, spectral shift, and spectroscopic data components of the temperature and tangent altitude errors are treated separately in order to allow the correct treatment of the systematic interaction with the gain, ILS and spectral shift components of the ozone retrieval errors and to disentangle their random and systematic components.

For some of the interfering species that are not jointly fitted we use retrievals of previous MIPAS data versions (V5) in the radiative transfer forward calculations, and hence can make use of their respective random error covariance matrices. They can
be considered as better representative for the actual conditions of the measurement than climatological mean values. For interfering species which are not available from previous data versions, perturbation spectra are calculated based on climatological data and on an assumption of the 1-$\sigma$ errors. In total, the contribution of interfering species to the ozone error budget is close to negligible due to the availability of many sufficiently strong ozone lines (see discussion in Sec. 4).

$H_2O$ uncertainties are not classified as parameter errors, since $H_2O$ mixing ratios are jointly fitted along with $O_3$.

Radiative transfer model errors are notoriously hard to quantify. Since no relevant malfunction of the radiative transfer model in use, KOPRA, (Stiller, 2000; Stiller et al., 2002), is known and the numerical accuracy parameters have been adjusted (Höpfner and Kellmann, 2000), we concentrate on the analysis of uncertainties in spectroscopic data. As already stated in Kiefer et al. (2021), uncertainties in spectroscopic data are not as clearly characterized as the retrieval scientist would like them to be. Band and transition-dependent uncertainty estimates are usually available and used in our error analysis. However, it is
often unclear what these uncertainty estimates represent, and information about error correlation is mostly missing. Since we use multiple ozone lines, it would be of utmost importance to know if the errors are fully random between the transitions and thus tend to cancel out, or if they are systematic and thus affect the retrieval in full, regardless how many lines are used.

For our error estimates we have made the following assumptions. Uncertainties in line intensities are parameterized according to a list by J.-M. Flaud and Ch. Piccolo (unpublished document, 2002), which roughly correspond to those of the
HITRAN16 data base (Gordon et al., 2017) and are reported as a function of band and rotational quantum number. For the stronger ozone lines the uncertainties are in the range of 1–2% (Table 2). For the broadening coefficients we use a parameteri-





**Table 2.** 1-$\sigma$ uncertainties of spectroscopic $O_3$ data as used for this work. $J$ and $K$ are the upper state rotational quantum numbers.

| Isotope | Band (HITRAN ID for vibr. levels) | Intensity Relative uncertainty | Broadening coeff. Relative uncertainty |
|---|---|---|---|
| $^{16}O^{16}O^{16}O$ | 2-1, 4-1, 5-1 | 0.02 (1 + $J$/70 + $K$/25) | 0.035 |
| | Other bands originating from the ground state (index = 1) | 0.03 (1 + $J$/60 + $K$/20) | 0.035 |
| | Bands originating from the lower states 2 ... 5 | 0.04 (1 + $J$/50 + $K$/17) | 0.075 |
| | Bands originating from the lower states 6 ... 14 | 0.06 (1 + $J$/40 + $K$/13) | 0.15 |
| | Bands originating from lower states > 14 | 0.10 (1 + $J$/35 + $K$/11) | 0.20 |
| Others | All | 0.03 (1 + $J$/60 + $K$/18) | 0.035 |

zation which roughly represents the uncertainties reported in HITRAN16. We use uncertainties of 3.5%, 7.5%, 15%, and 20%, depending on the respective ozone band, see Table 2.

We take all uncertainties to be characterizing 1-$\sigma$ uncertainties and make the most possible conservative assumption that the spectroscopic errors are fully correlated among the lines. We do not consider spectroscopic uncertainties of the interfering species. The reason is this. We use MIPAS-retrieved mixing ratios for these species which are affected by the spectroscopic errors in a way that the resulting signal (e.g. too high line intensity and accordingly too low mixing ratio) produces roughly the correct radiance signal. That is to say, the spectroscopic errors in interfering species and related errors in the mixing ratios cancel out and thus need no consideration in the ozone error budget.

As a rough guideline, spectroscopic uncertainties can be considered as relative errors. That is to say, to a first order they scale with the ozone mixing ratio.

Figure 2 shows the error budget for three selected atmospheric conditions, namely northern midlatitude summer day, tropical day, and southern polar winter night, for ozone retrievals from the FR (left column) and RR (right) measurement period. Tables and figures showing the error budgets of the remaining 31 representative atmospheres can be found in the supplement document.

The gain, instrument line shape (ILS), spectral shift, and spectroscopic data components of the temperature and tangent altitude errors are treated separately in order to allow the correct treatment of the systematic interaction with the gain, ILS and spectral shift components of the ozone retrieval errors and to disentangle their random and systematic components. However, for the plot the respective error pairs are quadratically added up and represented by one curve. That means, that, e.g., the orange lines in Fig. 2 represent the ozone retrieval errors that come directly from the ILS uncertainty, and those which are propagated via the T and LOS retrieval errors due to the ILS uncertainty.

It is evident that the spectroscopic error (blue open circle) is by far the single most important contributor to the total error in the altitude range 20–55 km for all three atmospheres. The noise error becomes large at altitudes where the ozone vmr is





low, and above 60 km begins to dominate the error budget of the RR ozone profiles. The impact of the low temperatures in the southern polar winter night stratosphere is well discernible, as the values increase to about 2% at 25–30 km while in the tropics and at northern midlatitudes they are at the 1% level. We included 21 interfering species in the calculation of the error budget. However, according to Fig. 2, where the total of all of these (calculated as square root of the sum of the respective variances) is shown with triangle symbols, the contribution in most altitudes is among the smallest ones. This is because ozone has strong lines in MIPAS A and AB bands, and it is rather easy to select enough microwindows with low to negligible signal of other gases.

As can already be seen from Fig. 2, there is no major inconsistency between the total errors (black diamonds) of FR and RR measurements for the three cases shown. A more detailed analysis shows that FR and RR total errors are quite close for most of the representative atmospheres in the altitude range 20–50 km. Above and below the FR data exhibits total errors which are a factor 2 (at 65 km) and 2–3 (at 10 km) greater than in the respective RR data. However, single error contributions can show quite a different behaviour. The error due to uncertainty in the spectral shift is 2–3 times larger in FR compared to RR at virtually all altitudes, albeit its contribution to the error budget is rather small. The ratio of FR to RR for the instrument ILS-induced errors is about one from 20–35 km, has a peak of 1.6 at 40 km, decreases towards one again at 50 km, and increases to values exceeding 3 above 60 km and below 15 km. A possible explanation is the difference in the spectral resolution: the influence on the exact shape of a spectral line is expected to be stronger for better spectral resolution. Since the ILS error has a significant contribution to the total error, the differences in FR and RR below 20 and above 50 km correspondingly contribute to the respective differences in the total error budget. The noise error of FR data is comparable to that of RR data at 20 km and decreases to only 2/3 of the latter at 45 km, while above 50 km and below 20 km ist is 1.5–2 times larger. Propagated errors from the preceding T+LOS retrieval, in general, are greater in FR than in RR data by a factor of approximately 1.5 above 20 km and 2–3 below. The ratio of the FR and RR errors due to uncertainties in spectroscopy, which already has been identified above as the main contributor to the total error, is close to unity between 20 and 50 km, reaches values of about 1.2 above and 2–3 below, and hence is, below 20 km, the main contributor to the differences in the total error between FR and RR data.

In the next sections we discuss the relevant measurement, parameter and model error sources, following the recommendations of TUNER (Towards Unified Error Reporting, von Clarmann et al., 2020). We categorize each error source either as chiefly random (Sections 4.1) or as chiefly systematic (4.2), depending on whether it causes mainly scatter or a bias. For mixed errors that cause both bias and scatter ("headache errors"), both components are reported separately. We aggregate the resulting random and systematic error components separately. All estimated errors are reported in terms of standard deviations ($1\sigma$). Averaging kernels are provided as additional information (Section 5.1).

## 4.1 Random errors

Following the recommendation of TUNER (von Clarmann et al., 2020), we define random errors as errors which explain the standard deviation of the differences between collocated measurements by two instruments measuring the same state variable. The main contributors to the total random error in the altitude range 20–50 km are, roughly ordered by their magnitude: measurement noise (0.8–3%), tangent altitude uncertainties (0.6–2%), the random component of gain calibration uncertainty



(however, sometimes being greater than the LOS-uncertainty), uncertainty in offset calibration, spectral shift uncertainty, and the uncertainties in those interfering species' abundances that are not changed in the retrieval, with the latter two components contributing usually far less than 1% each in this altitude range.

The state of the atmosphere can exert an influence on the size of the contribution of a specific error component to the budget. The contribution of, e.g., noise, is larger for colder and smaller for warmer atmospheres as is illustrated in Fig. 2 and discussed in Sec. 4. Error correlation matrices for the measurement noise error (not shown here) can be made available to data users on request.

## 4.2  Systematic errors

All errors that cause a long-term bias between measurements by different instruments are categorized as systematic errors. Since systematic errors are often modulated by the atmospheric state, the categorization of random versus systematic errors can only be approximative.

Chief contributors to the MIPAS systematic error budget are uncertainties in spectroscopic data with respect to the intensities and broadening coefficients of the used spectral lines, uncertainties in the MIPAS modulation efficiency that leads to uncertain-
ties in the ILS, and the persistent part of the gain calibration uncertainty, which is by far dominated by detector nonlinearity issues (see Table 3 of Kleinert et al., 2018).

## 5  Results

### 5.1  Averaging kernels and vertical resolution

Figure 3 shows examples of averaging kernel rows of the MIPAS ozone retrievals for sample geolocations of FR (a) and RR
(b) measurements. The vertical resolution of the ozone profiles is estimated as the full width at half maximum of the respective row of the averaging kernel matrix. All altitudes show higher peak values of the kernel rows for RR compared to FR and, since the norm of the kernel rows is one, this already hints at a better vertical resolution for RR ozone profiles.

Figure 4 depicts the vertical resolution over altitude along the orbit for orbits 4577 (FR, panel a) and 35800 (RR, panel b): indeed, the resolution is better for the RR data in all altitudes. The vertical resolution varies with the vertical scan pattern and,
in the case of RR measurements, with its latitudinal variation. The vertical fine structure of the data is caused by the fact that the retrieval grid is finer than the scan-/measurement grid. Local minima of the vertical resolution are reached, when a retrieval grid point lies close to or on a measurement grid point. However, we shall not look into details of this fine structure in what follows, but rather discuss the properties of the vertical resolution in the sense of altitude-smoothed profiles, averaged over time (e.g., over April 2003 for FR and 2009 for RR).

The vertical resolution for FR measurements starts at values of 2.5 km at the lowest altitudes and increases almost linearly to 6 km at 70 km altitude. RR measurements show values of 2 km at the lowest altitudes, which increase to 3 km at 20 km and stay constant up to 27 km. The vertical resolution increases linearly from 27 to 37 km up to a value of 4.2 km and stays





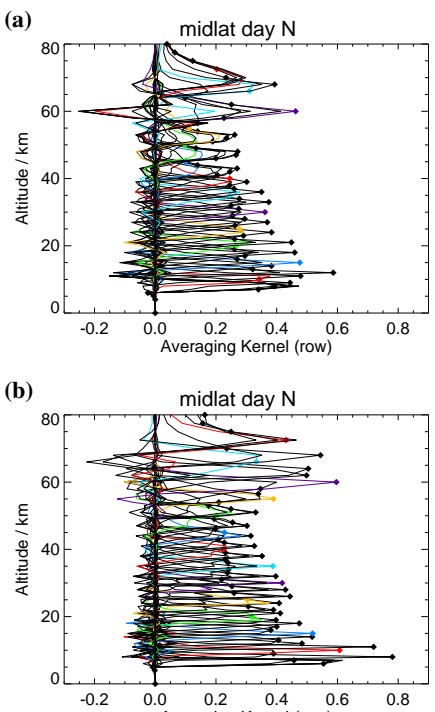

**Figure 3.** Rows of the averaging kernel matrix for ozone profiles recorded during the FR and RR measurement period. The rows are colored every 5 km up to 60 km. FR data (a) are taken from 14 Jan 2003 (Envisat orbit 4577) at 46.1°N, 155.1°W and solar elevation 17.6°; RR data (b) are from 4 Jan 2009 (orbit 35800) at 48.4°N, 116.4°E and solar elevation 14.6°.

constant further up to 50 km. Above, an increase to 5.5 km at 70 km is featured. However, it has to be noted that there is a latitude dependence of the RR scan grid (see Fig. 4b) which shifts these values in altitude. The region of reduced vertical
resolution discernible around 40 km and 60°N in Fig. 4b is caused by the very low temperatures (less than 200 K) which reduce atmospheric signal in the corresponding IR spectra.

The actual values of the vertical resolution of MIPAS $O_3$ are provided for each limb scan along with the result data set.

### 5.2 Horizontal averaging kernels and resolution

The horizontal resolution and information displacement were analyzed using the method of von Clarmann et al. (2009a). This
method exploits the information contained in the 2-dimensional averaging kernels. The relevant dimensions are altitude and the horizontal along-line-of-sight dimension. The horizontal smearing $r_{\mathrm{hor},z}$ at altitude $z$ is calculated as

$$r_{\mathrm{hor},z} = 2\sqrt{2\ln 2 \sum_l \frac{h_{z;l}(l-d_z)^2}{\sum_l h_{z;l}}}, \tag{1}$$

where $d_z$ is the information displacement (see below) at altitude $z$, and $h_{z;l}$ is the element of the horizontal information matrix at altitude $z$ that characterizes the horizontal gridpoint $l$. The latter is derived from the 2-D averaging kernel matrix by





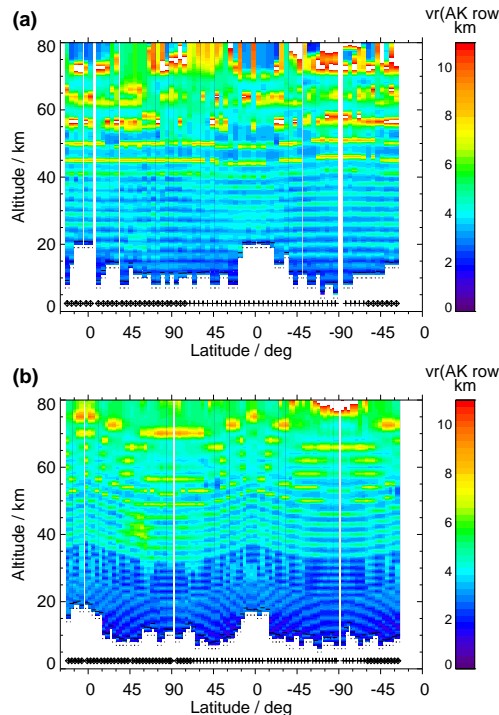

**Figure 4.** Vertical resolution calculated from the rows of the averaging kernel matrix for ozone profiles recorded during the FR (a) and RR (b) measurement period. Data for orbits 4577 (a) and 35800 (b) are shown, respectively. The symbols just above the latitude axis indicate the position of the measurement and give the illumination conditions (diamond: night; plus: day).

vertical integration of the absolute values of its entries. For horizontal averaging kernels resembling a Gaussian function, this measure of the horizontal smearing is equivalent to the full width at half maximum but it is by far more robust in the case of horizontal averaging kernels with two pronounced maxima. We report this quantity as 'horizontal smearing', not as 'horizontal resolution', because in cases of horizontal undersampling, where the horizontal smearing is smaller than the distance between two subsequent limb scans, the horizontal resolution is limited by the sampling, not by the horizontal information smearing. The horizontal information distribution has been analyzed for a limb scan at 47.9°S, 39.8°W, and 22° solar elevation, recorded during Envisat orbit 6075 on 29 April 2003 for FR measurements and a limb scan at 57.1°S, 176.1°W, and -32.5° solar elevation, recorded during Envisat orbit 39483 on 18 September 2009 for RR measurements.

For the FR measurements, the horizontal resolution is essentially sampling-limited. Only 40 km altitude the horizontal smearing occasionally slightly exceeds the horizontal sampling of MIPAS, i.e., the ground track distance of two subsequent limb scans, which is about 500 km. At most altitudes, the horizontal smearing is even considerably narrower (third column of Table 3). Also for the RR nominal measurement mode, the horizontal resolution is sampling-limited or in the same order of magnitude as the horizontal sampling of about 410 km for most altitudes. In our example only at 40, 66, and 70 km the RR smearing slightly exceeds the sampling (fifth column of Table 3).


**Table 3.** Horizontal information distribution for FR and RR.

|  | FR | FR | RR | RR |
| --- | --- | --- | --- | --- |
| Altitude | Displ. | Smearing | Displ. | Smearing |
| (km) | (km) | (km) | (km) | (km) |
| 5 | 177.8 | 266.8 | 123.1 | 263.2 |
| 10 | 158.6 | 375.5 | 130.2 | 316.7 |
| 15 | 131.3 | 394.5 | 103.4 | 327.5 |
| 20 | 94.5 | 349.5 | 66.5 | 312.8 |
| 25 | 61.9 | 356.0 | 28.0 | 297.9 |
| 30 | 24.1 | 351.7 | 17.5 | 345.0 |
| 35 | -15.8 | 340.5 | -14.7 | 325.9 |
| 40 | -43.5 | 436.4 | 1.7 | 465.3 |
| 45 | -58.4 | 541.3 | -52.2 | 341.3 |
| 50 | -119.8 | 423.6 | -66.2 | 342.0 |
| 55 | -46.8 | 582.0 | -72.6 | 330.3 |
| 60 | -98.5 | 492.6 | -77.0 | 347.9 |
| 66 | -91.8 | 528.2 | -39.4 | 426.7 |
| 70 | -121.4 | 549.6 | -50.1 | 502.3 |

The information displacement is the horizontal distance between the nominal geolocation of the measurement and the point

where most information comes from. The latter is calculated as the averaging-kernel-weighted mean horizontal coordinate. Our sign convention is such that displacements towards the satellite are associated with positive displacements. For both the FR and the RR measurements, the information displacement is considerably smaller than the horizontal sampling at all altitudes (second and fourth column of Table 3, respectively). Thus, we see no risk that the data are misinterpreted in terms of the geolocation they refer to. For the FR measurements, the information displacement is, broadly speaking, negative for altitudes

below 30 km and positive above. The information displacement of the RR measurements behaves just in the opposite way.

## 5.3   Ozone differences with respect to previous data versions

A comparison of daily means of ozone version 8 versus version 5 results for 20 February 2009 is shown in Figure 5. Most pronounced differences are seen over the North pole. While MIPAS retrievals are fairly independent of a priori information within the altitude range covered by observations, they do depend on assumptions on the atmospheric state above the uppermost

tangent altitude. In version 5 the temperature retrieval depended on climatological upper stratospheric and mesospheric temperature that did not represent the actual elevated stratopause on that day. The resulting version 5 temperature errors propagated on the ozone data product. In contrast, version 8 ozone relies on a temperature retrieval that used a priori data for the actual day (see Section 3.1) and is thus more adequate at altitudes where the measurement does not resolve the vertical temperature distribution. At an altitude of 55 km, resulting ozone differences reach 1 ppmv.





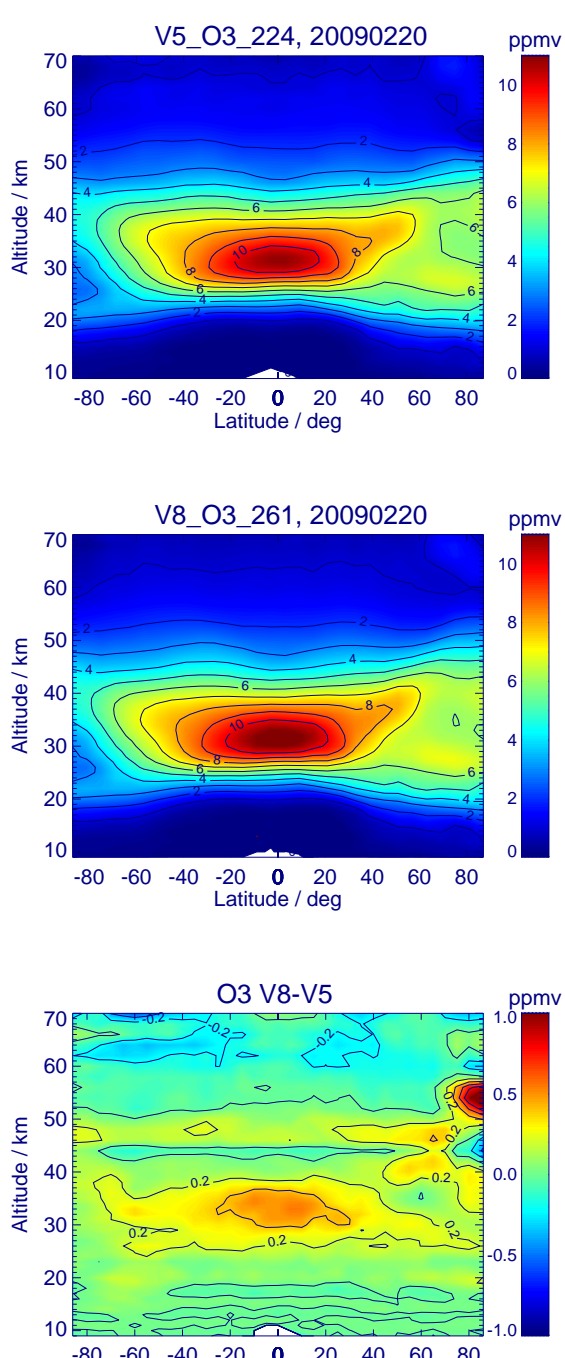

**Figure 5.** Global ozone distributions on 20 February 2009, MIPAS version V5_O3_224 (top panel) and version V8_O3_261 (middle panel). The lower panel shows the differences.



**(a)**

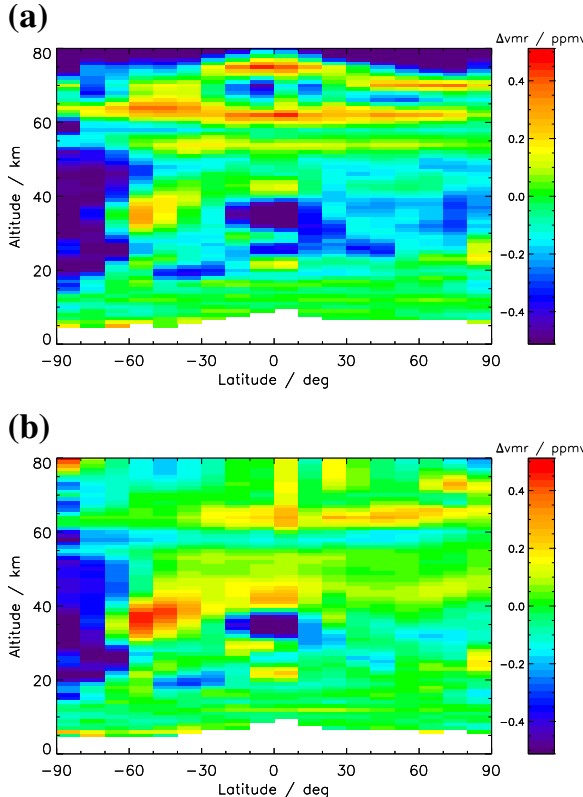

**(b)**

**Figure 6.** Differences between the means of all September data for RR and FR data. Top panel shows difference V5R_O3_224 minus V5H_O3_21, lower panel V8R_O3_261 minus V8H_O3_61. See text for details.

### 5.3.1 Consistency between full-resolution and reduced-resolution results

To assess the consistency of the results between the two measurement periods the difference between RR and FR monthly mean data was calculated for data versions V5 and V8 in 10° latitude bins. The monthly means are based on all available data. E.g., September means for the second measurement period data of each September in 2005–2012 were used. Figure 6 shows the respective differences (RR minus FR) for V5 data (top panel)and V8 data (bottom panel) for September. In the altitude range 0–70 km the structure of the differences is roughly the same. However, V8 differences between reduced and full resolution data seem to be more positive compared to the V5 differences. Above 70 km the patterns differ considerably, with the negative values in V5 indicating that at these altitudes there is a conspicuous inconsistency between reduced and full resolution results, which has disappeared in V8.

As a measure for the degree of consistency we use altitude/latitude means over the difference per month, and the corresponding standard deviation. All available latitudes and three different sets of altitude ranges are used. The respective data is shown in Fig. 7 for all months and the different altitude ranges. In the altitude range 20–50 km V5 and V8 data show essentially the same course of mean values, apart from a roughly constant bias. However, the majority of mean values of V8 differences data





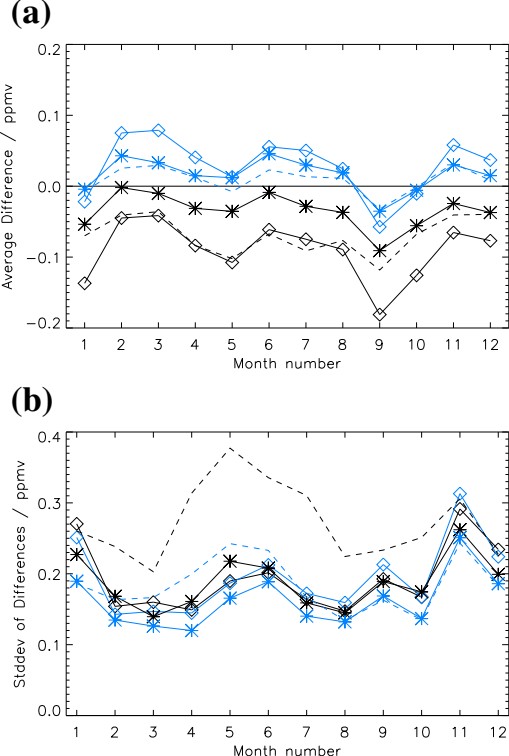

**Figure 7. (a)** Monthly mean differences between RR and FR ozone data. **(b)** Standard deviation of these differences. Differences and standard deviations are averaged over all latitudes and over altitude ranges 20–50 km (solid lines with diamonds), 0–70 km (solid lines with asterisk), and 0–80 km (broken lines). V5 data are shown with black, and V8 data with blue curves. See text for more details.

is closer to zero than in the corresponding V5 data, which shows purely negative values. This could be interpreted as both, a reduced step in ozone values between the full and reduced resolution periods, and a reduced (negative) drift in the data. The

latter interpretation is supported by Eckert et al. (2014), who state for V7 data that: "Drifts at 2-sigma significance level were mainly negative for ozone relative to Aura MLS and Odin OSIRIS and negative or near zero for most of the comparisons to lidar measurements." While there are some changes between the non-linearity correction for V7 and V8 data, the general impact with respect to data drift should be rather similar in both versions. In the altitude range 0–70 km (essentially the entire measurement range of MIPAS nominal mode measurements) the V5 data still exhibit purely negative values. Finally, mean

values for the altitude range 0–80 km are more negative again for V5 data, compared to the 0–70 km range, indicating that above 70 km the differences between reduced and full resolution values become strongly negative. This latter behaviour might well be attributed to the improved temperature retrieval (see Section 3.4 of Kiefer et al. (2021)). The overall picture, namely that the V8 curves are closer together compared to the V5 curves, together with the fact that the V8 curves are closer to zero than the corresponding V5 curves, shows that the consistency between RR and FR data is better for V8 data. Since there is

no hard proof, the final judgement has to be left to the ozone data validation. However, the curves of the means indicate a





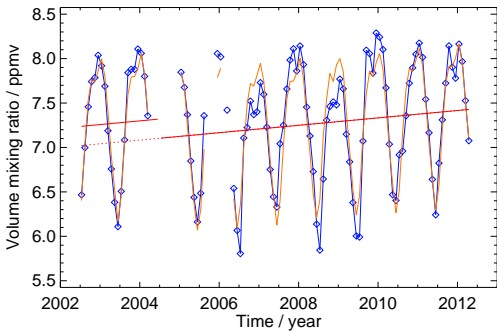

**Figure 8.** Fit of linear trend, yearly harmonics up to 2-month period, and step (arbitrarily assumed to occur July 2004) to an ozone monthly means time series. V8 monthly means at 30 km and in latitude range 40°S–50°S are blue with symbols while the fit is orange. Linear trend lines are red solid, the trend for the second period is extended into the first period as red dotted line to better illustrate the step between FR and RR measurement periods. In this example the fitted linear trend is 0.42 ppmv/decade and the fitted step is -0.22 ppmv.

behaviour which is consistent with what is expected as impact of several measures and improvements, and we value this as a hint that there is not only an improvement in general, but that there are two known reasons for this, namely the improved upper atmosphere V8 temperature data and the improved non-linearity correction.

The standard deviations presented in the lower panel of Fig. 7 show that in the 20–50 km altitude range (diamonds) there is
virtually no difference between V5 and V8. For 0–70 km the V8 data (asterisk) for all months exhibits slightly smaller values than V5. For the 0–80 km range (broken lines) there are considerable differences, with standard deviations of V8-differences being quite consistent with those of the other altitude ranges, which does not hold for V5 data. Again, this can be understood as an impact of the more realistic temperature at higher altitudes.

### 5.3.2   Temporal behaviour

Two aspects of the changes in temporal behaviour between MIPAS ozone data of versions V5 and V8 are in what follows: first, the step in ozone volume mixing ratio, which shows up in time series between FR and RR data, and second the drift with respect to data of satellites, LIDAR, and Ozonesondes, which has been analyzed by Eckert et al. (2014) and Laeng et al. (2018) for preceding ozone data versions. To assess these two quantities a suitable function is fitted to each time series at the retrieval altitudes and within a latitude bin of 10°. This function consists of a sum of terms for average, linear trend, and
yearly harmonics and sub-harmonics up to a period of two months. Additionally, a step between data before/after June 2004 is implemented (the exact separation date is not important, as long as it is between the FR and RR measurement periods). Two of the fit parameters are used to characterize the changes between V5 and V8 data: linear trend and step between FR and FR data. It has to be noted, however, that the fit parameter linear trend is not intended to give a reliable value of the ozone trend at the specific altitude and latitude band. If in the following we use the term *linear trend* we solely refer to our fit parameter. Ozone
trends will be called *(linear) ozone trends*. We employ neither QBO-terms, like e.g. Eckert et al. (2014) (see their Equation





**Table 4.** Mean values of the fit parameter step between FR and RR measurements and of the fit parameter linear trend (see text for details). The step in vmr is given in ppmv, the linear trend values in ppmv/year, and the decadal drift (linear trend in V8 minus linear trend in V5) in ppmv/decade. Values for the decadal drift MIPAS V5 minus MLS (v2.2 data) are calculated from the data which was used for Fig. 4 of Eckert et al. (2014) and which relies on more sophisticated methods for trend estimation than used in this work. Errors are standard errors of the mean.

|  | Altitude 35–45 km | Altitude 20–50 km |
| --- | --- | --- |
| step in V5 | -0.049±0.006 | -0.074±0.004 |
| step in V8 | 0.021±0.006 | -0.011±0.005 |
| linear trend in V5 | -0.0066±0.0011 | -0.0031±0.0008 |
| linear trend in V8 | 0.0069±0.0010 | 0.0060±0.0008 |
| decadal drift V8 - V5 | 0.1347±0.0150 | 0.0912±0.0113 |
| decadal drift MIPAS V5 - MLS | -0.1626±0.0048 | -0.0906±0.0036 |

1), nor sophisticated statistics since the only goal is to characterize the differences between V5 and V8. Detailed trend/drift analyses are beyond the scope of this work. Figure 8 illustrates the method for an altitude of 30 km and the latitude bin 40°S–50 °S.

We selected the altitude range 35–45 km, where MIPAS V5 ozone shows, on average, a clear negative drift compared to
both, MLS-Aura and OSIRIS (cf. Figs. 4 and 6 of Eckert et al. (2014)). This gives us 11x18 (altitude steps times latitude bins) time series and hence 191 values for each fitted parameter of the function. A second calculation was done for altitudes between 20 and 50 km to get an impression of how robust the results are if essentially the entire stratospheric data enters the procedure. From this procedure 558 single time series, and hence values per parameter, result. Mean values and standard errors of the mean are calculated from these two altitude ranges for the two parameters characterizing the vmr step and the linear trend.
The results are shown in Table 4. Clearly there is a significant reduction in the magnitude of the step in vmr between the two measurement periods for both altitude ranges. The decadal drift, i.e. the difference between the respective linear trends of V8 and V5 data in the altitude range 35–45 km is estimated to 0.1347± 0.0150 ppmv/decade. This average drift can be compared with, e.g., the average drift value for MIPAS minus MLS v2.2 of -0.1626±0.0048 ppmv/decade calculated from the data used as a basis for Figure 4 of Eckert et al. (2014). However, it has to be considered that the latter data rely on more sophisticated
methods for trend estimation than used in this work. Still, we think that comparing only differences of linear ozone trends with differences of our fitted linear trends is justified. There still is a discrepancy (note that the sign of either has to be reversed for direct comparison) but is is reduced by a large amount, roughly 80% of the initial drift between MIPAS and MLS would disappear changing from V5 to V8 retrieval results. In the altitude range 20–50 km the agreement between the decadal drifts calculated from the Eckert et al. data and our data is virtually perfect, i.e. 100% of the initial drift between MIPAS and MLS
would disappear. Actually, these results appear to be too good and not quite consistent with the data shown in Fig. 6 of Kiefer et al. (2021), where still a conspicuous residual drift of the V8 temperature data compared to microwave sounders seems to





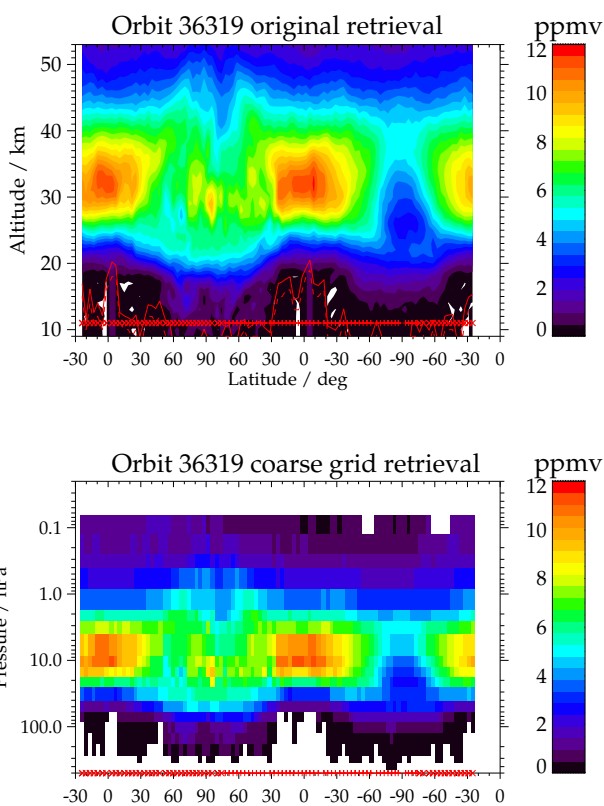

**Figure 9.** Ozone distribution for orbit 36319 on 20 February 2009 in the standard representation on the fine retrieval grid, where the vertical resolution is state- and thus time-dependent (upper panel), and on the coarse retrieval grid (lower panel), where the vertical resolution is defined by the vertical grid.

remain. One caveat, which might contribute to explain the good ozone drift reduction (with respect to MLS v.2.2) is that we use full monthly mean data while Eckert et al. (2014) work with collocated data. Still we see our results, based on preliminary estimations as presented above, as a strong indication that a substantial progress with respect to the drift problems found in

V5 data has been achieved. However, a thorough quantification of the ozone drift of MIPAS V8 ozone data still has to be performed.

# 6 Maximum likelihood representation

The ill-posedness of the inverse problem on the fine grid (see Section 3.2) requires regularization as described in Section 3.3. As a consequence, the averaging kernels deviate from unity. The averaging kernels, that characterize the content of a priori

information in the retrieval, depend on the atmospheric state and thus vary with time. For some applications, particularly the





analysis of time series, the resulting time-dependence of the vertical resolution poses non-trivial problems to the data user. Trend analysis (e.g., Yoon et al., 2013) or the analysis of annual cycles (e.g., Hegglin et al., 2013) belong to this category. Due to these problems, and since every statistics toolbox offers solutions to deal with varying errors, it appears desirable to offer an alternative representation of the data which is user-friendly in a sense that the data user does not need to care about averaging

kernels.

To that regard, our new data product includes both $O_3$ profiles retrieved with the original retrieval scheme, where the numerical constraint causes, via regularization, some content of *a priori* information in the product, and an alternative data set which is free of formal prior information (von Clarmann et al., 2015). The latter product is not meant to replace the regular product but is provided in parallel to offer the data user the choice between otherwise identical products. This additional data set is

henceforth called 'maximum likelihood product' (ML-product) even though this terminology is not fully appropriate because, depending on the constraint chosen, also the product obtained with the regularized retrieval can still have the epistemological nature of a maximum likelihood estimate, unless the regularization complies with the *maximum a posteriori* estimation scheme in a strong sense, i.e. a probability distribution can be assigned to the *a priori* distribution. In order to avoid unnecessary complication and to stay intuitive, we still use this simplified terminology.

Figure 9 shows results from Envisat orbit 36319 on 20 February 2009 from the retrieval on the fine grid, where the non-unity averaging kernel has to be considered (upper panel) and the coarse-grid, so called maximum-likelihood representation, where the altitude resolution is entirely defined by the vertical grid. For the convenience of modellers, a pressure grid has been chosen, whose gridpoints are commonly used in the modeling community, and thus avoids some unnecessary interpolation. The ozone data are provided as layer means and not as level values, i.e., the profile is transferred into a stepwise constant

profile while abundances integrated over the respective altitude range are retained. In particular, for the analysis of time series, this representation is advantageous, because there is no obvious straight-forward method to calculate meaningful time-series of data of varying vertical resolution.

## 7   Conclusions

MIPAS IMK/IAA ozone data presented in this work are based on the most recent version 8 level-1b spectra and were pro-
cessed using a retrieval approach improved over previous versions with respect to the temperature distributions used for the retrieval, the microwindow selection, the altitude-dependence of the regularization, the information on interfering species, the treatment of the radiance offset correction and background continuum, numerical issues, and the consideration of non-local thermodynamic equilibrium emission.

A TUNER-compliant uncertainty assessment is presented. In the altitude range 20–50 km the total error is below 10%.
Spectroscopic errors of ozone and carbon dioxide (propagated via the use of previously calculated T and LOS results) are the dominant components of the error budget. Errors from uncertainties of instrumental line shape function, gain calibration, and spectral noise also show significant contributions, while the error due to interfering gases is almost negligible. For most of the





cases the random error ranges between 2% and 5% in the altitude range 20–50 km, while the systematic error dominates the total error and is in the order of 6% to less than 10%.

The improved consistency of ozone data between the FR and RR measurement periods is demonstrated. A preliminary analysis of the drift issue indicates that a significant reduction of the drift found in version 5 data has been achieved.

As an alternative to the standard data product, a coarse-grid representation of the data is introduced and will be made publicly available. In this additional data set, the altitude resolution is defined solely by the vertical grid and thus is constant in time and invariant with respect to the atmospheric state.

While some of the known problems of preceding data versions have been solved by improvements of the level-1b data and the retrieval strategy, the use of version 8 level-1b data seems to have counteracted our effort to reduce the slight positive bias in the upper stratosphere. Only recently it has become known to us that self-inconsistency between ozone bands has been corrected in the HITRAN2020 data set (Gordon et al., 2021). This suggests that the spectroscopic data used in our study might contain line intensities too low by 1.8 to 2.8%, explaining a corresponding high bias in our ozone mixing ratios. A confirmation

of the drift reduction, as indicated by our analyses, requires comparisons with independent data. The quantitative analyses of both these issues are deferred to a dedicated validation study.

*Author contributions.*   MK developed the retrieval setup, coordinated and partly performed related test calculations and error estimation, and had the final editorial responsibility for this paper. TvC wrote large parts of the text, organized related discussions and cared about TUNER compliance of error estimates. BF provided the parameterized non-LTE approach, based on preceding work by MLP, MGC and himself. BF,

MGC and MLP took care that the retrieval setup was developed in a consistent way for nominal, middle, and upper atmospheric measurement modes. NG was responsible for spectroscopy issues, developed parts of the error estimation software, carried out some of the retrieval tests, and developed the coarse grid retrieval setup. UG provided and maintained the retrieval software and developed parts of the error estimation software. SK and ALinden run the retrievals. BF, SK and MH provided the horizontal averaging kernels. MH developed parts of the error estimation software. ALaeng contributed to quality control. GPS evaluated the preceding data versions for deficiencies to be removed in

version 8, organized the interfacing between IMK and IAA and took care of the quality control. All authors contributed to the development of the retrieval setup, discussed the results, and contributed to the final text.

*Competing interests.*   Some authors are members of the editorial board of journal AMT. The peer-review process was guided by an independent editor, and the authors have also no other competing interests to declare. The authors declare that they have no conflict of interest.

*Acknowledgements.*   This study was partly funded by DLR under contract no. 50EE1547 (SEREMISA). The IAA team was supported by

MCIU under projects PID2019-110689RB-100/AEI/10.13039/501100011033.

Spectra used for this work were provided by the European Space Agency. We would like to thank the MIPAS Quality Working Group for enlightening discussions, Claus Zehner for invaluable support, and Guido Levrini for motivating us to develop an independent MIPAS



data processor. The computations were done in the frame of a Bundesprojekt (grant MIPAS_V7) on the Cray XC40 "Hazel Hen" of the High-Performance Computing Center Stuttgart (HLRS) of the University of Stuttgart.



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
