# Peer review of "Version 8 IMK/IAA MIPAS ozone profiles: nominal observation mode"

_Atmospheric Measurement Techniques, 2022_

## Author Response (AR1)

**Response to the Referee comments for manuscript Nr. amt-2022-257**

**General remarks**

1. In parallel to the reviewing process we found a bug in the software which is used to calculate the error contributions. The program has been corrected and all error calculations have been redone. Accordingly, the plots shown in Figure 2, and all plots and tables in the supplement, have been changed. However, the differences to the error data shown in the initial manuscript are very small, and have an impact only on minor error contributions. No changes of the respective text/discussions were necessary.

2. The text of the original manuscript contained an error with respect to the sign of the horizontal information displacement which is corrected now. In the same section the statement about the altitudes where the horizontal smearing exceeds the horizontal sampling of MIPAS is made more precise (see marked up file).

**Replies to the Referee Comments, and Performed Actions**

Original questions/comments of the referees are marked by **RC:** and set in *italic font.*

**Replies to Referee #1**

**RC:** *This paper presents a new version of the MIPAS stratospheric ozone profile data set. The work is well motivated by changes in the retrieval strategy, which will be important for some community users of the data set, and by the new calibration of the level-1b spectra. The paper is quite detailed although some elements listed below are difficult to follow for those outside the project. The work fits well within the scope of AMT and should be published after consideration of the minor points below. It is recommended that the abstract be rewritten as it is currently quite lengthy and should be focused on the most important points of the new dataset.*
**Reply:** As both referees unanimously raise this issue, we have rewritten the abstract in a more concise manner, leaving out statements which might be too detailed, and concentrating on the major methodical and input data changes and on the major findings.
**Text Changes:** The abstract has been condensed and partly rephrased (see marked up document).

**RC:** *Section 3.1 on the preceding temperature retrieval is not clear. What is actually done in this step? Who are "these authors"? A better (not longer) summary of this step from Kiefer et al 2021 is needed as it comes up several times in following sections.*
**Reply:** We agree to restructure and rewrite this section for clarity

**Text Changes:** The respective text has been restructured and rewritten, see the marked up file (note that the respective section now is 4.1).

**RC:** *Can the authors provide any insight about why the inclusion of horizontal gradient did not work?*
**Reply:** Unfortunately, we are not very clear on this issue. We know from the analysis of the horizontal averaging kernels that the horizontal range to which MIPAS is sensitive is only a few hundred km (typically below 400–500 km, according to the measurement mode). We suspect that MIPAS is not very sensitive to ozone variations along such a short distance. In exceptional cases (e.g. vortex edge) there might be larger horizontal ozone variations to which MIPAS might indeed be sensitive, but in such situations the parametrization of the horizontal structure by a simple gradient would be inadequate to represent the true horizontal structure, and the retrieval of additional parameters to describe the horizontal variability would destabilize the retrieval.
**Text Changes:** Any text on this subject would be rather speculative. Hence, we keep the respective section of the manuscript as it is.

**RC:** *Why is it better to jointly fit water vapor instead of using the regular product? Explain what is meant by "In order to avoid propagation of assumptions on the water vapour profiles on the ozone data product".*
**Reply:** Our retrievals are done sequentially. After shift correction there is the T+LOS retrieval, which is followed directly by the ozone retrieval. Hence, there is no regular product of V8 water vapour available at the time of the ozone retrieval. Still, V5 water vapour data could be used. However, the ozone spectral lines used for the retrieval are appreciably interfered with water vapour spectral lines. The spectroscopic data of these lines may be inconsistent with the spectroscopic data used for the V5 water vapour retrieval in a different spectral band. Thus, the spectral signal caused by the water vapour lines in the ozone microwindows might not accurately be accounted for if the water vapour concentrations from the water vapour retrieval are used. To avoid propagation of related errors on the ozone results, water vapour is fitted jointly with ozone. We will add this information to the manuscript.
**Text Changes:** The section on water vapour (3.2.3 in the original manuscript, now 4.2.3) has been extended for clarification of this issue (see marked up manuscript).

**RC:** *It would be useful to connect the weakening of the smoothing regularization to impact on vertical resolution.*
**Reply:** Yes, that seems very reasonable.
**Text Changes:** In Section 4.3 (3.3 in the submitted version) the sentence: "This modification allowed to weaken the Tikhonov-type smoothing regularization at all altitudes by approximately 20%." was complemented by another sentence to read now: "This

modification allowed to weaken the Tikhonov-type smoothing regularization at all altitudes by approximately 20%. This, in turn, improved the vertical resolution of the resulting ozone profiles." Section 6.1 (formerly 5.1) contains a new paragraph with details of the improvements in the vertical resolution for FR and RR ozone profiles.

**RC:** *Numerical issues: what is meant by the "result flipping back and forth". This statement needs more rigor and explanation.*
**Reply:** More detailed text similar to that in Sect. 3.12 of the MIPAS V8 temperature retrieval paper by Kiefer et al. [1] has been implanted in the manuscript.
**Text Changes:** The new text, replacing parts of lines 182–183 is:
"In the retrieval code an 'oscillation detection' has been activated which identifies failure of convergence in the sense that the iteration flips back and forth between two minima of the cost function according to $\vec{x}_{i+1} \approx \vec{x}_{i-1}$ and $\vec{x}_i \approx \vec{x}_{i-2}$. In this case $\vec{x}_{i+1}$ is set to $\frac{\vec{x}_{i+1} + \vec{x}_i}{2}$, and further iteration steps are performed."

**RC:** *What causes the strange arching shapes in the noise-component in Fig 1? Is this due to the mismatch between retrieval and measurement grids?*
**Reply:** We assume that the referee refers to the structures mainly visible below 20 km in Fig. 1b (RR data). In general, the mismatch between retrieval an measurement grid are causing the stripes. This is well visible in Fig. 1a (FR data) for virtually all altitudes: If the retrieval grid is finer than the tangent altitude grid, typically the errors due to noise are larger where the tangent altitude coincides with the retrieval altitude, because the bulk of information comes from one measurement only. If the retrieval altitude is between two tangent altitudes, both provide information, thus the error is smaller. Conversely, the altitude resolution is better in the first case and coarser in the second case. This effect causes oscillations both in the error profiles and the vertical resolution profiles. When the tangent altitude pattern changes as a function of latitude as in RR measurements, while the retrieval grid remains constant, we get arches as observed in the plots. In the figure below, the circles represent the latitude dependence of the tangent altitudes, the horizintal lines represent the retrieval grid, and the dots highlight the position where the retrieval grid coincides with a tangent altitude. These dots form arches as observed in the plots.

[Figure]

**Text Changes:** None.

**RC:** *The discussion of spectroscopic errors and the overall link to the TUNER recommendations is appreciated.*
**Reply:** Thank you!
**Text Changes:** None.

**RC:** *Including a plot of the vertical resolution with the cases shown in Fig 3 would be useful.*
**Reply:** A good suggestion.
**Text Changes:** Figure 3 has been complemented by plots of the vertical resolutions. The text of Section 6.1 (formerly 5.1) referring to Figure 3 has been changed accordingly.

**RC:** *The discussion on the "improvement" shown in Fig 5 due to reduction of temperature propagation errors from high altitudes sounds good. Is it verifiable that v8 is in fact better?*
**Reply:** We know that nominal retrieval results do better coincide with middle atmosphere results (which use measurements up to 100 km) for V8 data than for V5 data. This leads us to believe that V8 ozone is superior to V5 data at high altitudes. Currently dedicated ozone validation studies are under way. Looking at northern polar winter conditions will be a part of these.
**Text Changes:** None.

**RC:** *Should the new maximum likelihood representation for the analysis of the drift? Why not?*
**Reply:** We assume the question is whether the new maximum likelihood representation should be used for the assessment of the instrument drift. As pointed out in the first paragraph of section 7 (formerly Sec. 6), these new data are expected to be free of time-dependent effects in the vertical resolution and hence better suited for, e.g., trend studies. So, when comparing to other instruments in this respect, which could be used as a way to assess the residual drift, the new representation clearly should be preferred. Inter-version comparisons, e.g. V8 versus V5 will not be possible, since yet there is no maximum likelihood representation for V5 data.
**Text Changes:** None.

**Replies to Referee #2**

**RC:** *The paper by Kiefer et al. reports on a new version (V8) of ozone retrievals from the MIPAS satellite dataset. Most importantly, the new retrievals are driven by a new calibration version of the underlying spectra. Essentially, the paper provides a reference document to the community what the changes are wrt. to previous MIPAS ozone retrieval versions and how performance has improved. These improvements are particularly important for the usage of MIPAS data in trend analyses and studies of the upper atmosphere. The paper is very well written and all the analyses are rigorous. Therefore, I do not have any but a few technical comments.*

**RC:** *Abstract: I would support reviewer 1 in requesting the abstract to be rewritten in a more concise way highlighting the most important findings.*
**Reply:** Agreed.
**Text Changes:** The abstract has been condensed and partly rephrased.

**RC:** *l.5: temperatures → temperature data*
*l.8: Ozone lines → ozone absorption lines*
*l.39: mod ededicated → mode dedicated*
l.79: As the cause... rephrase → An inadequate... was identified to be the cause ...
l.181: was set to → was set to a spacing of
l.236: to allow the correct treatment → to allow for the correct treatment
l.313: not changed in the retrieval → not varied in the retrieval
l.436: was done → was carried out

**Reply:** These corrections and wording suggestions are well justified.
**Text Changes:** The above corrections and wording suggestions have been implemented in the revised manuscript.

**RC:** *l.53: There is no section 1.2. Is a section header "1.1" really required?*
**Reply:** We see the point.
**Text Changes:** Subsection 1.1 has been promoted to become a full section (Sec. 2).

**RC:** *l105: The preceding temperature retrieval → The general reader might be confused by what "preceding" refers to. Why not just "Temperature retrieval"?*
**Reply:** "Preceding" is meant to allude to the fact that we do the retrievals of temperature and the diverse gases in a sequence, one after another. Since this is already briefly mentioned in Sec. 3, l.93f of the original manuscript (now Sec. 4, l.88f) we prefer to leave it as it is.
**Text Changes:** None

**References**

[1] Kiefer, M., von Clarmann, T., Funke, B., García-Comas, M., Glatthor, N., Grabowski, U., Kellmann, S., Kleinert, A., Laeng, A., Linden, A., López-Puertas, M., Marsh, D., and Stiller, G. P.: IMK/IAA MIPAS temperature retrieval version 8: nominal measurements, Atmos. Meas. Tech., 14, 4111–4138, https://doi.org/10.5194/amt-14-4111-2021, 2021.